# OpenReview forum: "A Geometric Lens on LLM Abilities through Joint Embedding Item Response Theory"
_TMLR — Accepted by TMLR_

### Review · Reviewer_uqzN · 2026-03-28

**Summary Of Contributions:**

The paper „A Geometric Lens on LLM Abilities through Joint Embedding Item Response Theory looks at the performance of LLMs in benchmark through the lens of IRT, i.e., tries to model how difficult tasks are and how this relates to the likelihood that the LLMs give the correct answer. The idea is to use the two embeddings to model the ability and difficulty, i.e., the embeddings of the questions and a learned model embedding.

**Audience:**

Yes

**Audience Explanation:**

The general topic is interesting, though the framing and future application should be clarified.

**Claims And Evidence:**

No

**Claims Explanation:**

Unfortunately, the paper is extremely hard to follow, making it very hard for me to judge the actual contribution. The core problem is that the concepts that are used are not really defined in the main body of the paper and that the definitions themselves leave room for interpretation.
For example, how E_Q and E_M are computed is never defined in the main body of the paper. For Equation (2) they are just defined as embeddings in a shared embedding space. To understand where the model embedding E_M comes from, one must dig deep into the Appendix to see that this are in fact the parameters of a logistic regression model that tries to predict correctness of answers based on the embeddings.
Further, the definition of the question embeddings is also unclear to me. From my understanding, the function f_{base} is the LLM embedding, but I do not really understand what the adapter g_\theta is used for and how it is trained. I also could not find this anywhere. My guess is that this is somehow trained using the tasks, possibly as (part of) a prediction head, but I am not fully clear on this.
Arguably, these are the two most important concepts in the whole paper and for both the definition is fuzzy. The paper should be rewritten to make the main body self-contained with all core information required to follow the notations.

Second, I am not really sure what I should take from the results, given my (possibly incomplete or wrong) understanding of the method. It seems like the paper suggests that we can learn a logistic regression which can be interpreted as embedding vector, such that the cosine-similarity of this learned embedding with that of a query to a LLM can be used as indicator for the likelihood that a question is answered correctly.
The claim is that this works well for all tasks that are similar to those that were considered during the training of the IRT-model. However, the paper never really dives into this. Importantly, whenever the analysis shows weaknesses of this approach (e.g., for MMLU) this is attributed to the diversity of the task. For me, this implies that the method only works well, if the overall task diversity considered is bounded, likely because the model embeddings are not sufficiently expressive to model answer correctness in general. This makes me wonder what this at the end of the day is:

-	a method to analyze how well models are likely handling within and out-of-distribution data for a large, diverse range of tasks or
-	a method to analyze how well models are likely handling within and out-of-distribution data for a narrow range of tasks without too much internal diversity.

The framing implies the former, the empirical results imply the later. Note, that many aspects in the analysis are indirectly affected by this, e.g., everything on geometric alignment via embedding orientation (which does not really make sense for diverse data, as the authors themselves write), the importance of leave-one-out benchmarks, etc.

Regarding the propositions: While they make sense, I suggest to rather move them completely to the Appendix to make room for a better explanation of the method itself. The reason is that they are obvious for embeddings. I would argue that the propositions are rather axioms that were postulated and a method was chosen to create an IRT that fulfills these axioms. In other words, the propositions are not a consequence of the design, but the design was chosen such that these (reasonable) propositions are fulfilled.

I also wonder about the k-means approach to cluster the embedding space. There is nothing on the metric that is used here. Is this done using an Euclidean distance? That would not really make sense in the embedding space, right? Cosine similarity would not take the magnitude into account, which the authors deem to be important. Scaled-dot product similarity could be an option to instead take this into account. This needs to be clarified, as otherwise it is unclear if the cluster analysis is meaningful. There should also be information about how good the clustering is, e.g., how well-separated the clusters are.

**Requested Changes:**

- Improve the readability, especially by providing clear definitions for all core concepts.
- Clarify the actual utility of the approach in practically relevant scenarios and revisit how this works with respect to task diversity.
- Better explain the clustering approach and explain what the usage of the selected metric implies for the outcome, as well as analyze how good the produced clusters fit the data.
- Consider framing the propositions as axioms that must be fulfilled by a reasonable IRT for LLMs.

---

> ### Author Response · Authors · 2026-04-27
> **Response to Reviewer**
>
> We thank the reviewer for the detailed and constructive feedback. We have reorganized the comments into individual points below for clarity, and address each with the corresponding modifications in the revision.
>
> >How are $E_Q$, $E_M$ and the JE-IRT training objective defined, and is the method now self-contained in the main text?
>
> We have rewritten the Model Structure paragraph in Section 3 (highlighted in brown) to make all core definitions self-contained. The question embedding $E_Q = g_\theta(f_{\mathrm{base}}(Q))$ is now explicitly defined as a frozen pretrained text encoder (e.g., ModernBERT-Large, which is separate from the LLMs being evaluated) followed by a trainable two-layer MLP adapter. The model embedding $E_M = T[i]$ is explicitly defined as a learnable vector from a trainable embedding table, learned solely from observed response patterns rather than derived from model internals. The training data (binary correctness matrix $\{y_{i,j}\}$) and the loss function (binary cross-entropy) are now stated with an explicit equation in the main text.
>
> >Is JE-IRT merely logistic regression on embeddings, or a geometric generalization of IRT with additional structure?
>
> JE-IRT is a geometric generalization of IRT. The central idea is to use the embedding norm to represent difficulty and the embedding direction to encode semantics, so that correctness is determined by the interaction between a model's projected ability along a question's direction and the question's difficulty. This geometric structure is what distinguishes JE-IRT from generic prediction models and is what enables the analyses in Sections 4.4--4.6: directional alignment predicts OOD behavior, norms track difficulty, and linear probes recover cross-subject abilities.
>
> Here we used logit link because our task is binary prediction. As we show in Appendix H, the framework can be generalized to other tasks beyond binary prediction and thus with different link functions than logit function.
>
> >What does the MMLU result imply about task diversity, and does it limit the method or only the single-direction aggregate analysis?
>
> The MMLU result limits the mean-direction analysis, not the method itself. When MMLU is included in training, JE-IRT predicts its questions accurately (the ``All'' bars in Figure 3). The OOD drop when MMLU is held out reflects that the remaining benchmarks do not cover MMLU's diverse directions, and the directional alignment metric correctly predicts this. The issue is that summarizing MMLU with a single mean direction is too coarse for a benchmark spanning 57 subjects, as confirmed by its wide angular dispersion in the kernel PCA projection (Appendix E). We have revised the discussion in Section 4.4 (highlighted in brown) to clarify this distinction.
>
> >How have the propositions and clustering analysis been clarified, including their role, metric choice, and cluster quality?
>
> We have reframed the propositions as desirable properties that JE-IRT is designed to satisfy, rather than presenting them as unexpected consequences. (highlighted in brown in Section 3)
>
> For the clustering analysis, we now state explicitly that k-means is applied to unit-normalized embeddings, so that Euclidean distance reduces to cosine dissimilarity. We clarified this in Section 4.6 as highlighted in brown.
>
>  We have also added silhouette scores to assess intrinsic cluster separation and included a stability analysis in Appendix J (highlighted in brown) showing all metrics across varying $K$ (37--87) and random seeds. The consistently low silhouette scores indicate that the embedding space does not decompose into sharply separated discrete clusters, which is consistent with our finding that LLM abilities are distributed across overlapping directions rather than concentrated in distinct subject-specific regions.

---

> > ### Comment · Reviewer_uqzN · 2026-05-10
> >
> > Thank you for this detailed response and many improvements to the manuscripts.
> >
> > With respect to the MMLU vector, the value proposition is still not clear to me. It still seems like a clear limitation that should be acknowledged: the method is not suitable to capture the diversity of complex benchmarks. I still do not see any clear argument or data that this is not the case and the method is suitable to predict task performance on data sets in general. I strongly suggest to either provide clear data or acknowledge the limitation and frame the problem space, for which this can be used reliably, accordingly.

---

> > > ### Author Response · Authors · 2026-05-10
> > >
> > > Thank you for your comments. JE-IRT should not be interpreted as guaranteeing reliable extrapolation to arbitrary unseen domains. This is a general consideration for any data-driven evaluation framework: estimating a model's ability along a given direction requires training evidence for that ability. Therefore, the variation in generalization performance across completely held-out benchmarks reflects the coverage of the response-matrix training set, rather than an assumption that all unseen benchmarks can be predicted equally well.
> > >
> > > At the same time, the geometric structure of JE-IRT makes this coverage issue explicit. By decomposing abilities into directional components, the framework helps identify which ability directions are represented in the training benchmarks and which are weakly covered. We have revised the manuscript and uploaded a new version to clarify this point. Specifically, we added a clarification on page 9, in the paragraph immediately before Section 4.5, highlighted in brown. We also expanded the Limitations section, also highlighted in brown, to emphasize that broad benchmark coverage is important when applying JE-IRT to unseen domains.

---

### Review · Reviewer_SMeT · 2026-04-19

**Summary Of Contributions:**

This paper studies the evaluation of a given LLM on different tasks from the perspective of embeddings. Specifically, this paper introduces JE-IRT, a geometric item-response Theory, which embeds both the LLMs and the questions in a shared space. For the embedding in this space: the direction encodes the semantics and the norm encodes the question difficulty. The probability of correctness is determined by a projection-based interaction between the embeddings of the model and the question. On the empirical side, the authors show that the classical IRT fails; and the results of JE-IRT framework well aligns with the testing accuracy. In summary, it proposes a new framework to evaluate the models, which can help to predict the performance of the models on unseen tasks.

**Additional Comments:**

Questions:
- The JE-IRT framework assumes the difficulty is fully captured by the embedding norm. How robust is this assumption? Can the difficulty also have directional components?
- The model uses projection onto question direction as ability. Is it possible to consider other interactions/linkage? For instance, can we use others, e.g., bilinear functions, or cosine, or even other non-linear functions like kernels?
- The classical IRT is shown to fail. Can the authors elaborate, Is JE-IRT fundamentally a generalization of IRT, or closer to representation learning with a logistic head?

**Audience:**

Yes

**Audience Explanation:**

The evaluation of LLMs is important. This paper provides a representation-based way to characterize the performance of an LLM on a given task. It should be of interest to the community.

**Broader Impact Concerns:**

None.

**Claims And Evidence:**

Yes

**Claims Explanation:**

- The motivation is well explained and also supported by empirical results on the classical IRT methods.
- The interpretation is clear and is aligned with the intuitions: the direction represents semantics and the norm represents difficulty.
- The experiments corroborate the JE-IRT framework and showcase its effectiveness. It also studies the effect of the dimension on the prediction accuracy.

**Requested Changes:**

Weaknesses:
- The generalization is not that novel. As the classical IRT operates on the scalar parameters, it is kind of natural to extend it to the embedding space, where the angles and norms can be made meaningful by training an embedding encoder.
- The theoretical contributions are relatively weak. The propositions, regarding no global ordering, the stability bounds, are not surprising to me, since it operates on the vectors.
- While norm correlates with difficulty, AUC (0.73–0.77) is moderate.

---

> ### Author Response · Authors · 2026-04-27
> **Response to Reviewer**
>
> Thank you for your detailed review and helpful comments. We repsond point-by-point and incorporate the suggested clarifications in the revision.
>
> >The generalization is not that novel. As the classical IRT operates on the scalar parameters, it is kind of natural to extend it to the embedding space, where the angles and norms can be made meaningful by training an embedding encoder.
> >The classical IRT is shown to fail. Can the authors elaborate, Is JE-IRT fundamentally a generalization of IRT, or closer to representation learning with a logistic head?
>
> The contribution of this work is not purely algebraic. Traditional IRT treats item parameters as free per-question scalars learned from response patterns alone, with no connection to question content. A generic representation-learning model with a logistic head would learn a black-box predictor with no constrained geometric meaning. JE-IRT sits between these two: it ties question parameters to content through a text encoder, and constrains the geometry so that direction encodes semantics and norm encodes difficulty. This specific structure is what enables the interpretive analyses that constitute the core contribution of the paper: directional alignment predicting OOD generalization (Section 4.4), norms tracking difficulty (Section 4.5), clustering revealing an LLM-internal taxonomy distinct from human-defined subjects (Section 4.6), and linear probes recovering cross-subject abilities like the arithmetic axis (Section 4.6). None of these analyses would be possible with either traditional IRT or an unconstrained representation model.
>
> >While norm correlates with difficulty, AUC (0.73–0.77) is moderate.
>
> We have revised Section 4.5 (highlighted in purple) to clarify this point. The AUC of $0.73$--$0.77$ is computed using the embedding norm alone, which captures model-agnostic intrinsic difficulty. The full JE-IRT prediction relies on both norm and directional interaction between model and question embeddings. As shown in Section 4.1, there is no universal total ordering of models across questions, so correctness is inherently model-dependent and cannot be fully predicted by a single scalar. The moderate AUC is therefore expected and consistent with our framework: it confirms that the norm encodes a substantial difficulty signal, while the remaining variance is captured by the directional component.
>
> >The JE-IRT framework assumes the difficulty is fully captured by the embedding norm. How robust is this assumption? Can the difficulty also have directional components?
>
> We appreciate this question. In JE-IRT, the probability of correctness is determined by the difference between a model's projected ability along the question direction and the question's embedding norm (Eq. 3). Along a given direction, the norm fully captures difficulty by definition: a larger norm requires greater projected ability to answer correctly. However, discussing difficulty without specifying the direction is not meaningful, because different models have different projected abilities along different directions. This is the core motivation of JE-IRT: difficulty and topic are entangled in practice, and our framework captures this by modeling their interaction rather than treating difficulty as a single global scalar. Empirically, the norm alone achieves AUC 0.73-0.77 in predicting question difficulty (Section 4.5), confirming that it carries a strong difficulty signal while also reflecting that direction plays a complementary role that the norm alone cannot account for.
>
> >The model uses projection onto question direction as ability. Is it possible to consider other interactions/linkage? For instance, can we use others, e.g., bilinear functions, or cosine, or even other non-linear functions like kernels?
>
>
> Alternative interaction functions are possible in principle. For example, EmbedLLM uses a bilinear form, and cosine similarity or kernel-based interactions could also be used. However, each of these comes at the cost of interpretability. Cosine similarity discards the norm, which would remove our notion of difficulty. A bilinear form introduces a learned interaction matrix that entangles direction and magnitude, making it harder to attribute meaning to individual geometric components. The projection-based interaction in JE-IRT is specifically chosen because it cleanly separates the roles of direction (semantics) and norm (difficulty), which is what enables the geometric analyses central to our paper. We discuss this design choice in more detail in Appendix H.

---

### Review · Reviewer_MAYN · 2026-04-26

**Summary Of Contributions:**

This paper proposes JE-IRT, a joint embedding item response theory framework for LLM evaluation. Instead of modeling each LLM with a single scalar ability, JE-IRT embeds both models and questions in a shared geometric space, where a model’s projected ability along a question direction predicts correctness and the question norm is interpreted as difficulty. The paper first shows that traditional scalar IRT assumptions fail for LLMs, since model performance does not obey a universal total ordering across questions. It then demonstrates that JE-IRT achieves reasonable item-level correctness prediction, supports efficient integration of new LLMs by fitting only a model embedding, and yields interpretable question-side geometry such as difficulty norms, benchmark alignment, and cross-subject ability directions.

The main strengths are that the proposed framework is conceptually elegant, addresses a real limitation of aggregate-score evaluation, and provides an interesting geometric perspective on model-question interactions. The empirical analyses on question norms, OOD benchmark alignment, and arithmetic directions are also useful and suggest that such embeddings could support more fine-grained evaluation beyond leaderboard accuracy.

The main weaknesses are that the predictive baselines are incomplete, the broader evaluation value is not fully demonstrated, and the model-side geometry is under-explored. In particular, simple baselines such as majority prediction, per-model accuracy, per-question difficulty, and model/question bias models are important for interpreting the reported accuracy. The paper also mainly analyzes question embeddings, while model embedding norms, model-model similarities, and task-level ability vectors could provide stronger evidence that the learned space is a useful evaluation framework rather than only a correctness predictor.

**Additional Comments:**

This is an interesting paper with a promising conceptual framing. I would encourage the authors to develop the model-side and task-vector analyses more fully, since these would make the learned embedding space much more compelling as a new evaluation paradigm beyond aggregate accuracy.

**Audience:**

Yes

**Audience Explanation:**

Researchers working on LLM evaluation, benchmark design, model routing, and capability profiling would likely find this paper interesting. The idea of replacing scalar leaderboard rankings with a shared model-question embedding space is timely and potentially useful. Even if the current analysis is not yet fully developed, the framework offers a promising direction for representing model abilities and question difficulty in a more structured way.

**Broader Impact Concerns:**

I do not see major negative broader impact concerns. The main concern is that such a framework could be over-interpreted as providing definitive capability measurements, even though the embeddings depend on the benchmark distribution and correctness labels used to train them. The paper should emphasize that JE-IRT provides a data-dependent evaluation lens rather than an absolute measure of model ability.

**Claims And Evidence:**

Yes

**Claims Explanation:**

The paper provides convincing evidence for several core claims: traditional scalar IRT is insufficient for LLM evaluation, JE-IRT can predict model-question correctness, and question embedding norms correlate with difficulty. The experiments on new-model integration and OOD benchmark alignment also support the claim that the learned space captures some transferable structure.

**Requested Changes:**

1. Add simple but important predictive baselines. One question I care about is how much better JE-IRT performs compared to simple heuristic baselines. The paper should report baselines such as:
- uniform random guessing whether a model answers a question correctly or not;
- per-model majority prediction, i.e., for each model, always predicting whether that model usually answers questions correctly or incorrectly based on its training-set accuracy;
- per-question majority prediction, i.e., for each question, predicting whether most models answer that question correctly or incorrectly; and
These baselines are necessary to interpret whether JE-IRT’s accuracy reflects genuine model-question interaction modeling rather than label imbalance, model strength, or question difficulty alone.

2. Strengthen the analysis of model-side geometry. The paper currently analyzes question directions and norms, but says much less about model embeddings. It would be valuable to study whether model norm correlates with overall capability, whether model-model similarity reflects model family/training recipe, and whether model embeddings move systematically with scale or post-training.
3. Demonstrate broader evaluation utility. To support the claim that JE-IRT is more than a prediction model, the paper should show at least one downstream use case, such as model routing, benchmark item selection, or capability diagnosis. A task-level vector analysis would be especially useful: for example, constructing math/medical/logical ability directions and testing whether model projections onto these directions predict held-out task performance.
4. Analyze stability of learned axes and clusters. The arithmetic-axis and clustering results are interesting but should be tested across random seeds, encoders, and data splits to establish that they are robust rather than post-hoc artifacts.

---

> ### Author Response · Authors · 2026-04-27
> **Response to Reviewer - 1/2**
>
> Thank you for the thoughtful feedback and suggestions. We appreciate the reviewer's engagement and address the main points below.
>
> >Add simple but important predictive baselines. One question I care about is how much better JE-IRT performs compared to simple heuristic baselines. The paper should report baselines such as:
>
> We have added three heuristic baselines to Section 4.2 and Figure 2 (highlighted in blue): overall majority vote ($63.40\%$), per-model majority vote ($63.59\%$), and per-benchmark majority vote ($71.09\%$). All baselines are computed on the training split and evaluated on the test split to avoid label leakage. JE-IRT surpasses all baselines at every embedding dimension tested.
>
> Regarding per-question majority vote, the test split consists of unseen questions that do not appear in training, so per-question statistics are unavailable at test time without introducing leakage. For this reason, we use per-benchmark majority as the strongest non-leaking proxy for question difficulty. This design reflects our goal: JE-IRT is intended to generalize to new questions via learned embeddings rather than rely on memorized per-question patterns.
>
> >Strengthen the analysis of model-side geometry. The paper currently analyzes question directions and norms, but says much less about model embeddings. It would be valuable to study whether model norm correlates with overall capability, whether model-model similarity reflects model family/training recipe, and whether model embeddings move systematically with scale or post-training.
>
> We investigated model-side geometry as suggested. In these preliminary analyses, we did not find a simple scalar or family-level organization: model embedding norm does not predict item-level correctness (AUC 0.499), which is expected since the norm is constant across all questions for a given model, so item-level ability is captured by the directional projection onto each question. We also observe that model norms are similar in magnitude across models.
>
> Mean within-family and across-family cosine similarities are both small across embedding dimensions. This is broadly consistent with the high effective-rank analysis already reported in Appendix E (Figure 7), suggesting that model-side variation may be distributed across multiple dimensions rather than captured by a single capability score or simple provider-family grouping.
>
>
> >Demonstrate broader evaluation utility. To support the claim that JE-IRT is more than a prediction model, the paper should show at least one downstream use case, such as model routing, benchmark item selection, or capability diagnosis. A task-level vector analysis would be especially useful: for example, constructing math/medical/logical ability directions and testing whether model projections onto these directions predict held-out task performance.
>
>
> We have added a cost-free routing experiment in Section 5 (section title highlighted in blue). For each test question, JE-IRT routes to the model with the highest predicted $P(\text{correct})$ using only the question text. JE-IRT achieves 65.17\% routing accuracy, improving over the best-average-model heuristic (55.58\%) by approximately 10 percentage points. It also surpasses the per-benchmark best heuristic (61.95\%), which requires knowing which benchmark each question belongs to, information unavailable when routing arbitrary queries in practice. Cost-aware routing, which balances accuracy against inference cost, requires additional modeling assumptions orthogonal to our framework, and we leave this for future work. More broadly, the analyses in Sections 4.4--4.6 further demonstrate the utility of the learned geometry: directional alignment predicts OOD behavior, norms track difficulty, and ability probes reveal structure not visible from aggregate scores or human-defined categories.
>
>
> >Analyze stability of learned axes and clusters. The arithmetic-axis and clustering results are interesting but should be tested across random seeds, encoders, and data splits to establish that they are robust rather than post-hoc artifacts.
>
> The difference-of-means probe used to construct the arithmetic axis is deterministic given the embeddings, so it introduces no additional source of instability. The embeddings themselves do depend on training initialization, but all key results are consistent across two base encoders with entirely different architectures, indicating that the geometric structure is robust. For clustering, we have added a stability analysis in Appendix J (highlighted) showing results across random seeds and varying numbers of clusters from 37 to 87. The variation across seeds is small and does not affect the conclusions.

---

> ### Author Response · Authors · 2026-04-27
> **Response to Reviewer - 2/2**
>
> >The main concern is that such a framework could be over-interpreted as providing definitive capability measurements, even though the embeddings depend on the benchmark distribution and correctness labels used to train them. The paper should emphasize that JE-IRT provides a data-dependent evaluation lens rather than an absolute measure of model ability.
>
> We agree with this concern and have added a paragraph (highlighted in blue) to the Limitations section clarifying that JE-IRT provides a data-dependent evaluation lens rather than an absolute measure of model ability.

---

### Decision · Action_Editor_L46d · 2026-06-09

**Recommendation:** Accept as is

**Audience:**

Yes

**Audience Explanation:**

Yes. This is an interesting paper which addresses the important topic of evaluating LLM abilities.

**Claims And Evidence:**

Yes

**Claims Explanation:**

Yes. This is a good paper with substantial evaluations to support its claims. While one reviewer seems to remain a bit concerned about the explanation of the performance drop on MMLU, I have double checked the revised discussion and the experiments in Appendix E, and feel that the authors have appropriately answered the concerns (i.e., "The OOD drop when MMLU is held out reflects that the remaining benchmarks do not cover MMLU's diverse directions, and the directional alignment metric correctly predicts this. The issue is that summarizing MMLU with a single mean direction is too coarse for a benchmark spanning 57 subjects, as confirmed by its wide angular dispersion in the kernel PCA projection (Appendix E).").